# Visual Interaction Networks: Learning a Physics Simulator from Video

**Nicholas Watters,  Andrea Tacchetti,  Théophane Weber**
**Razvan Pascanu,  Peter Battaglia,  Daniel Zoran**
DeepMind
London, United Kingdom
```
{nwatters, atacchet, theophane,
razp, peterbattaglia, danielzoran}@google.com
```

## Abstract

From just a glance, humans can make rich predictions about the future of a wide range of physical systems. On the other hand, modern approaches from engineering, robotics, and graphics are often restricted to narrow domains or require information about the underlying state. We introduce the *Visual Interaction Network*, a general-purpose model for learning the dynamics of a physical system from raw visual observations. Our model consists of a perceptual front-end based on convolutional neural networks and a dynamics predictor based on interaction networks. Through joint training, the perceptual front-end learns to parse a dynamic visual scene into a set of factored latent object representations. The dynamics predictor learns to roll these states forward in time by computing their interactions, producing a predicted physical trajectory of arbitrary length. We found that from just six input video frames the Visual Interaction Network can generate accurate future trajectories of hundreds of time steps on a wide range of physical systems. Our model can also be applied to scenes with invisible objects, inferring their future states from their effects on the visible objects, and can implicitly infer the unknown mass of objects. This work opens new opportunities for model-based decision-making and planning from raw sensory observations in complex physical environments.

## 1 Introduction

Physical reasoning is a core domain of human knowledge [22] and among the earliest topics in AI [24, 25]. However, we still do not have a system for physical reasoning that can approach the abilities of even a young child. A key obstacle is that we lack a general-purpose mechanism for making physical predictions about the future from sensory observations of the present. Overcoming this challenge will help close the gap between human and machine performance on important classes of behavior that depend on physical reasoning, such as model-based decision-making [3], physical inference [13], and counterfactual reasoning [10, 11].

We introduce the Visual Interaction Network (VIN), a general-purpose model for predicting future physical states from video data. The VIN is learnable and can be trained from supervised data sequences which consist of input image frames and target object state values. It can learn to approximate a range of different physical systems which involve interacting entities by implicitly internalizing the rules necessary for simulating their dynamics and interactions.

The VIN model is comprised of two main components: a visual encoder based on convolutional neural networks (CNNs) [17], and a recurrent neural network (RNN) with an interaction network (IN) [2] as its core, for making iterated physical predictions. Using this architecture we are able to learn a model which infers object states and can make accurate predictions about these states in future time

steps. We show that this model outperforms various baselines and can generate compelling future rollout trajectories.

## 1.1 Related work

One approach to learning physical reasoning is to train models to make state-to-state predictions. One early algorithm using this approach was the "NeuroAnimator" [12], which was able to simulate articulated bodies. Ladicky et al. [16] proposed a learned model for simulating fluid dynamics based on regression forests. Battaglia et al. [2] introduced a general-purpose learnable physics engine, termed an Interaction Network (IN), which could learn to predict gravitational systems, rigid body dynamics, and mass-spring systems. Chang et al. [7] introduced a similar model in parallel that could likewise predict rigid body dynamics.

Another class of approaches learn to predict summary physical judgments and produce simple actions from images. There have been several efforts [18, 19] which used CNN-based models to predict whether a stack of blocks would fall. Mottaghi et al. [20, 21] predicted coarse, image-space motion trajectories of objects in real images. Several efforts [4, 6, 26, 27] have fit the parameters of Newtonian mechanics equations to systems depicted in images and videos, though the dynamic equations themselves were not learned. Agrawal et al. [1] trained a system that learns to move objects by poking.

A third class of methods [5, 8, 9, 23], like our Visual Interaction Network, have been used to predict future state descriptions from pixels. However, in contrast to the Visual Interaction Network, these models have to be tailored to the particular physical domain of interest, are only effective over a few time steps, or use side information such as object locations and physical constraints at test time.

## 2 Model

The Visual Interaction Network (VIN) learns to produce future trajectories of objects in a physical system from video frames of that system. The VIN is depicted in Figure 1, and consists of the following components:

- The **visual encoder** takes a triplet of frames as input and outputs a state code. A state code is a list of vectors, one for each object in the scene. Each of these vectors is a distributed representation of the position and velocity of its corresponding object. We apply the encoder in a sliding window over a sequence of frames, producing a sequence of state codes. See Section 2.1 and Figure 2a for details.

- The **dynamics predictor** takes a sequence of state codes (output from a visual encoder applied in a sliding-window manner to a sequence of frames) and predicts a candidate state code for the next frame. The dynamics predictor is comprised of several interaction-net cores, each taking input at a different temporal offset and producing candidate state codes. These candidates are aggregated by an MLP to produce a predicted state code for the next frame. See Section 2.2 and Figure 2b for details.

- The **state decoder** converts a state code to a state. A state is a list of each object's position/velocity vector. The training targets for the system are ground truth states. See Section 2.3 for details.

## 2.1 Visual Encoder

The visual encoder is a CNN that produces a state code from a sequence of 3 images. It has a frame pair encoder $E_{\text{pair}}$ shown in Figure 2a which takes a pair of consecutive frames and outputs a candidate state code. This frame pair encoder is applied to both consecutive pairs of frames in a sequence of 3 frames. The two resulting candidate state codes are aggregated by a shared MLP applied to the concatenation of each pair of slots. The result is an encoded state code. $E_{\text{pair}}$ itself applies a CNN with two different kernel sizes to a channel-stacked pair of frames, appends constant $x, y$ coordinate channels, and applies a CNN with alternating convolutional and max-pooling layers until unit width and height. The resulting matrix of shape $1 \times 1 \times (N_{\text{object}} \cdot L_{\text{code}})$ is reshaped into a state code of shape $N_{\text{object}} \times L_{\text{code}}$, where $N_{\text{object}}$ is the number of objects in the scene and $L_{\text{code}}$ is the length of each state code slot. The two state codes are fed into an MLP to produce the final

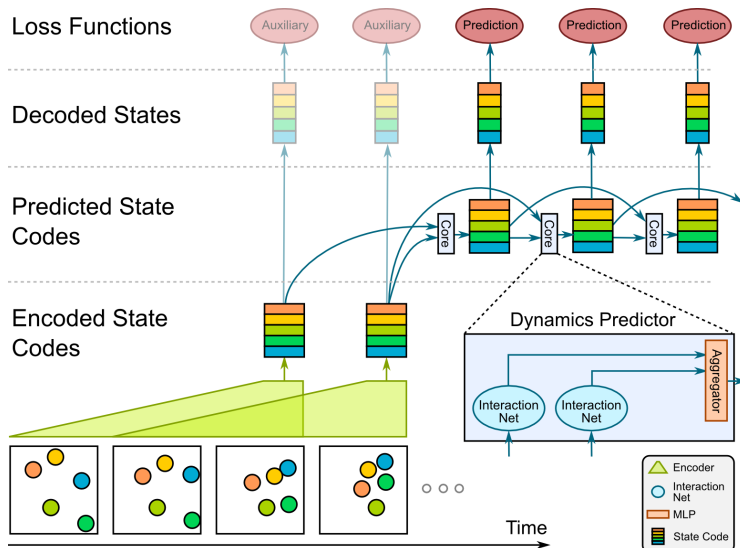

Figure 1: **Visual Interaction Network:** The general architecture is depicted here (see legend on the bottom right). The visual encoder takes triplets of consecutive frames and produces a state code for the third frame in each triplet. The visual encoder is applied in a sliding window over the input sequence to produce a sequence of state codes. Auxiliary losses applied to the decoded output of the encoder help in training. The state code sequence is then fed into the dynamics predictor which has several Interaction Net cores (2 in this example) working on different temporal offsets. The outputs of these Interaction Nets are then fed into an aggregator to produce the prediction for the next time step. The core is applied in a sliding window manner as depicted in the figure. The predicted state codes are linearly decoded and are used in the prediction loss when training.

encoder output from the triplet. See the Supplementary Material for further details of the visual encoder model.

One important feature of this visual encoder architecture is its weight sharing given by applying the same $E_{pair}$ on both pairs of frames, which approximates a temporal convolution over the input sequence. Another important feature is the inclusion of constant coordinate channels (an x- and y-coordinate `meshgrid` over the image), which allows positions to be incorporated throughout much of the processing. Without the coordinate channels, such a convolutional architecture would have to infer position from the boundaries of the image, a more challenging task.

## 2.2 Dynamics Predictor

The dynamics predictor is a variant of an Interaction Network (IN) [2]. An IN, summarized in Figure 2b, is a state-to-state physical predictor model that uses a shared relation net on pairs of objects as well as shared self-dynamics and global affector nets to predict per-object dynamics. The main difference between our predictor and a vanilla IN is aggregation over multiple temporal offsets. Our predictor has a set of temporal offsets (in practice we use $\{1, 2, 4\}$), with one IN core for each. Given an input state code sequence, for each offset $t$ a separate IN core computes a candidate predicted state code from the input state code at index $t$. An MLP aggregator transforms the list of candidate state codes into a predicted state code. This aggregator is applied independently to the concatenation over candidate state codes of each slot and is shared across slots to enforce some consistency of object representations. See the Supplementary Material for further details of the dynamics predictor model.

As with the visual encoder, we explored many dynamics predictor architectures (some of which we compare as baselines below). The temporal offset aggregation of this architecture enhances its power by allowing it to accommodate both fast and slow movements by different objects within a sequence of frames. See the Supplementary Material for an exploration of the importance of temporal offset aggregation. The factorized representation of INs, which allows efficient learning of interactions even in scenes with many objects, is an important contributor to our predictor architecture's performance.

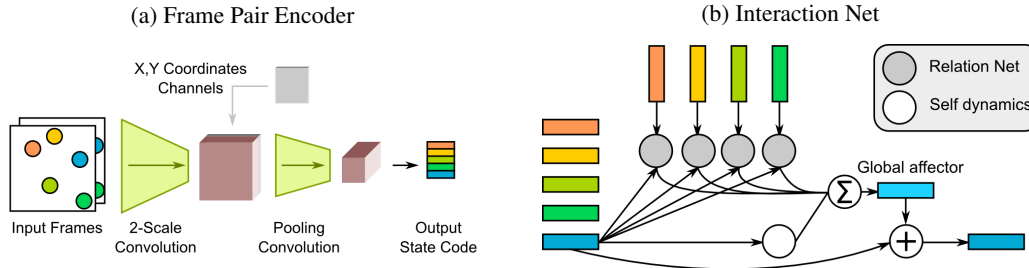

Figure 2: **Frame Pair Encoder and Interaction Net. (a)** The frame pair encoder is a CNN which transforms two consecutive input frame into a state code. Important features are the concatenation of coordinate channels before pooling to unit width and height. The pooled output is reshaped into a state code. **(b)** An Interaction Net (IN) is used for each temporal offset by the dynamics predictor. For each slot, a relation net is applied to the slot's concatenation with each other slot. A self-dynamics net is applied to the slot itself. Both of these results are summed and post-processed by the affector to produce the predicted slot.

## 2.3   State Decoder

The state decoder is simply a linear layer with input size $L_{code}$ and output size 4 (for a position/velocity vector). This linear layer is applied independently to each slot of the state code. We explored more complicated architectures, but this yielded the best performance. The state decoder is applied to both encoded state codes (for auxiliary encoding loss) and predicted state codes (for prediction loss).

## 3   Experiments

### 3.1   Physical Systems Simulations

We focused on five types of physical systems with high dynamic complexity but low visual complexity, namely 2-dimensional simulations of colored objects on natural-image backgrounds interacting with a variety of forces (see the Supplementary Material for details). In each system the force law is applied pair-wise to all objects and all objects have the same mass and density unless otherwise stated.

- **Spring** Each pair of objects has an invisible spring connection with non-zero equilibrium. All springs share the same equilibrium and Hooke's constant.
- **Gravity** Objects are massive and obey Newton's Law of gravity.
- **Billiards** No long-distance forces are present, but the billiards bounce off each other and off the boundaries of the field of vision.
- **Magnetic Billiards** All billiards are positively charged, so instead of bouncing, they repel each other according to Coulomb's Law. They still bounce off the boundaries.
- **Drift** No forces of any kind are present. Objects drift with their initial velocities.

These systems include previously studied gravitational and billiards systems [3, 1] with the added challenge of natural image backgrounds. For example videos of these systems, see the Supplementary Material or visit (https://goo.gl/yVQbUa).

One limitation of the above systems is that the positions, masses, and radii of all objects are either visually observable in every frame or global constants. Furthermore, while occlusion is allowed, the objects have the same radius so total occlusion never occurs. In contrast, systems with hidden quantities that influence dynamics abound in the real world. To mimic this, we explored a few challenging additional systems:

- **Springs with Invisibility**. In each simulation a random object is not rendered. In this way a model must infer the location of the invisible object from its effects on the other objects.
- **Springs** and **Billiards with Variable Mass**. In each simulation, each object's radius is randomly generated. This not only causes total occlusion (in the Spring system), but density is held constant, so a model must determine each object's mass from its radius.

To simulate each system, we initialized the position and velocity of each ball randomly and used a physics engine to simulate the resulting dynamics. See the Supplementary Material for more details. To generate video data, we rendered the system state on top of a CIFAR-10 natural image background. The background was randomized between simulations. Importantly, we rendered the objects with 15-fold anti-aliasing so the visual encoder could learn to distinguish object positions much more finely than pixel resolution, as evident by the visual encoder accuracy described in Section 4.1.

For each system we generated a dataset with 3 objects and a dataset with 6 objects. Each dataset had a training set of $2.5 \cdot 10^5$ simulations and a test set of $2.5 \cdot 10^4$ simulations, with each simulation 64 frames long. Since we trained on sequences of 14 frames, this ensures we had more than $1 \cdot 10^7$ training samples with distinct dynamics. We rendered natural image backgrounds online from separate training and testing CIFAR-10 sets.

## 3.2 Baseline Models

We compared the VIN to a suite of baseline and competitor models, including ablation experiments. For each model, we performed hyperparameter sweeps across all datasets and choose the hyperparameter set with the lowest average test loss.

The **Visual RNN** has the same visual encoder as the VIN, but the core of its dynamics predictor core is an MLP instead of an IN. Each state code is flattened before being passed to the dynamics predictor. The dynamics predictor is still treated as a recurrent network with temporal offset aggregation, but the dynamics predictor no longer supports the factorized representation of the IN core. Without the weight-sharing of the IN, this model is forced to learn the same force law for each pair of objects, which is not scalable as the object number increases.

The **Visual LSTM** has the same visual encoder as the VIN, but its dynamics predictor is an LSTM [14] with MLP pre- and post-processors. It has no temporal offset aggregation, since the LSTM implicitly integrates temporal information through state updates. During rollouts, the output state code from the post-processor MLP is fed into the pre-processor MLP.

The **VIN Without Relations** is an ablation modification of the VIN. The only difference between this and the VIN is an omitted relation network in the dynamics predictor cores. Note that there is still ample opportunity to compute relations between objects (both in the visual encoder and the dynamics predictor's temporal offset aggregator), just not specifically through the relation network. Note that we performed a second ablation experiment to isolate the effect of temporal offset aggregation. See the Supplementary Material for details.

The **Vision With Ground-Truth Dynamics** model uses a visual encoder and a miniature version of the dynamics predictor to predict not the next-step state but the current-step state (i.e. the state corresponding to the last observed frame). Since this predicts static dynamics, we did not train it on rollouts. However, when testing, we fed the static state estimation into a ground-truth physics engine to generate rollouts. This model is not a fair comparison to the other models because it does not learn dynamics. It serves instead as a performance bound imposed by the visual encoder. We normalized our results by the performance of this model, as described in Section 4.

All models described above learn state from *pixels*. However, we also trained two baselines with privileged information: **IN from State** and **LSTM from State** models, which have the IN and LSTM dynamics predictors, but make their predictions directly from state to state. Hence, they do not have a visual encoder but instead have access to the ground truth states for observed frames. These, in combination with the Vision with Ground Truth Dynamics, allowed us to comprehensively test our model in part and in full.

## 3.3 Training procedure

Our goal was for the models to accurately predict physical dynamics into the future. As shown in Figure 1, the VIN lends itself well to long-term predictions because the dynamics predictor can be treated as a recurrent net and rolled out on state codes. We trained the model to predict a sequence of 8 consecutive unseen future states from 6 frames of input video. Our prediction loss was a normalized weighted sum of the corresponding 8 error terms. The sum was weighted by a discount factor that started at $0.0$ and approached $1.0$ throughout training, so at the start of training the model must only predict the first unseen state and at the end it must predict an average of all 8 future states. Our

training loss was the sum of this prediction loss and an auxiliary encoding loss, as indicated in Figure 1. The model was trained by backpropagation with an Adam optimizer [15]. See the Supplementary Material for full training parameters.

## 4   Results

Our results show that the VIN predicts dynamics accurately, outperforming baselines on all datasets (see Figures 3 and 4). It is scalable, can accommodate forces with a variety of strengths and distance ranges, and can infer visually unobservable quantities (invisible object location) from dynamics. Our model also generates long rollout sequences that are both visually plausible and similar to a ground-truth physics, even outperforming state-of-the-art state-to-state models on this measure.

### 4.1   Inverse Normalized Loss

We evaluated the performance of each model with the Inverse Normalized Loss, defined as $L_{bound}/L_{model}$. Here $L_{bound}$ is the test loss of the Vision with Ground Truth Dynamics and $L_{model}$ is the test loss of the model in question (See Section 3.3). We used this metric because it is much more interpretable than $L_{model}$ itself. The Vision with Ground Truth Dynamics produces the best possible predictions given the visual encoder's error, so the Inverse Normalized Loss always lies in $[0, 1]$, where a value closer to $1.0$ indicates better performance. The visual encoder learned position predictions accurate to within 0.15% of the framewidth (0.048 times the pixel width), so we have no concerns about the accuracy of the Vision with Ground Truth Dynamics.

Figure 3 shows the Inverse Normalized Loss on all test datasets after $3 \cdot 10^5$ training steps. The VIN out-performs all baselines on nearly all systems. The only baseline with comparable performance is the VIN Without Relations on Drift, which matches the VIN's performance. This makes sense, because the objects do not interact in the Drift system, so the relation net should be unnecessary.

Of particular note is the performance of the VIN on the invisible dataset (spring system with random invisible object), where its performance is comparable to the fully visible 3-object Spring system. It can locate the invisible object's position to within 4% of the frame width (1.3 times the pixel width) for the first 8 rollout steps.

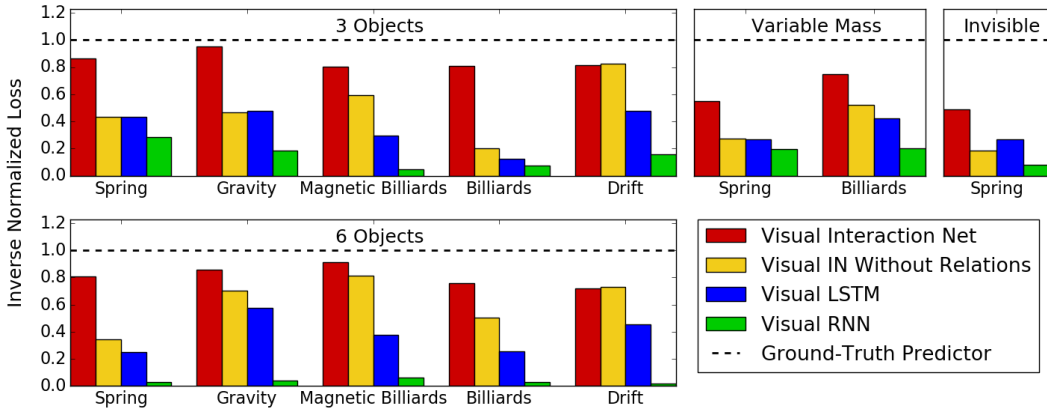

Figure 3: **Performance.** We compare our model's Inverse Normalized Loss to that of the baselines on all test datasets. 3-object dataset are on the upper row, and 6-object datasets are on the lower row. By definition of the Inverse Normalized Loss, all values are in $[0, 1]$ with $1.0$ being the performance of a ground-truth simulator given the visual encoder. The VIN (red) outperforms every baseline on every dataset (except the VIN Without Relations on Drift, the system with no object interactions).

### 4.2   Euclidean Prediction Error of Rollout Positions

One important desirable feature of a physical predictor is the ability to extrapolate from a short input video. We addressed this by comparing performance of all models on long rollout sequences and observing the Euclidean Prediction Error. To compute the Euclidean Prediction Error from a

predicted state and ground-truth state, we calculated the mean over objects of the Euclidean norm between the predicted and true position vectors.

We computed the Euclidean Prediction Error at each step over a 50-timestep rollout sequence. Figure 4 shows the average of this quantity over all 3-object test datasets with respect to both timestep and object distance traveled. The VIN out-performs all other models, including the IN from state and LSTM from state even though they have access to privileged information. This demonstrates the remarkable robustness and generalization power of the VIN. We hypothesize that it outperforms state-to-state models in part because its dynamics predictor must tolerate visual encoder noise during training. This noise-robustness translates to rollouts, where the dynamics predictor remains accurate even as its predictions deviate from true physical dynamics. The state-to-state methods are not trained on noisy state inputs, so they exhibit poorer generalization. See the Supplementary Material for a dataset-specific quantification of these results.

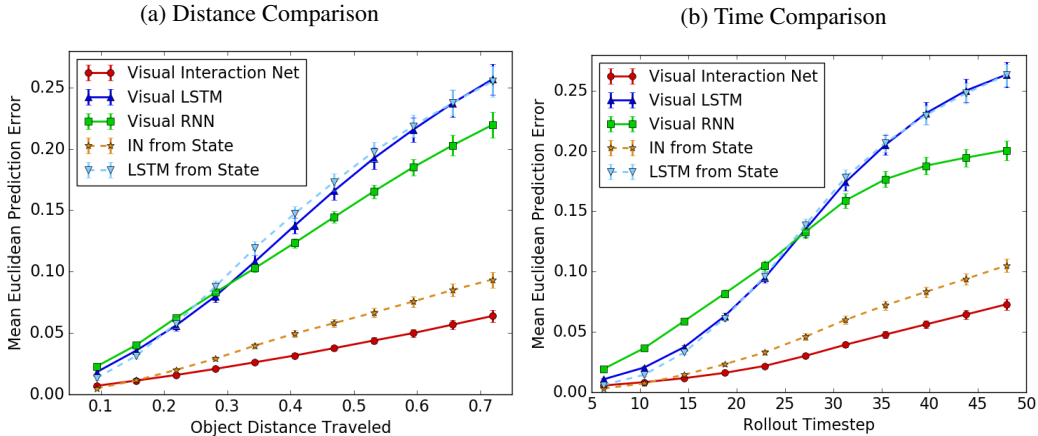

Figure 4: **Euclidean Prediction Error on 3-object datasets.** We compute the mean over all test datasets of the Euclidean Prediction Error for 50-timestep rollouts. The VIN outperforms all other pixel-to-state models (solid lines) and state-to-state models (dashed lines). Errorbars show 95% confidence intervals. **(a)** Mean Euclidean Prediction Error with respect to object distance traveled (measured as a fraction of the frame-width). The VIN is accurate to within 6% after objects have traversed 0.72 times the framewidth. **(b)** Mean Euclidean Prediction Error with respect to timestep. The VIN is accurate to within 7.5% after 50 timesteps. The optimal information-less predictor (predicting all objects to be at the frame's center) has an error of 37%, higher than all models.

### 4.3 Visualized Rollouts

To qualitatively evaluate the plausibility of the VIN's rollout predictions, we generated videos by rendering the rollout predictions. These are best seen in video format, though we show them in trajectory-trail images here as well. The backgrounds made trajectory-trails difficult to see, so we masked the background (only for rendering purposes). Trajectory trails are shown for rollouts between 40 and 60 time steps, depending on the dataset.

We encourage the reader to view the videos at (https://goo.gl/RjE3ey). Those include the CIFAR backgrounds and show very long rollouts of up to 200 timesteps, which demonstrate the VIN's extremely realistic predictions. We find no reason to doubt that the predictions would continue to be visually realistic (if not exactly tracking the ground-truth simulator) ad infinitum.

## 5   Conclusion

Here we introduced the Visual Interaction Network and showed that it can infer the physical states of multiple objects from video input and make accurate predictions about their future trajectories. The model uses a CNN-based visual encoder to obtain accurate measurements of object states in the scene. The model also harnesses the prediction abilities and relational computation of Interaction Networks, providing accurate predictions far into the future. We have demonstrated that our model performs well on a variety of physical systems and is robust to visual complexity and partially observable data.

Table 1: **Rollout Trajectories.** For each of our datasets, we show a sample frame, an example true future trajectory, and a corresponding predicted rollout trajectory (for 40-60 frames, depending on the dataset). The left half shows the 3-object regime and the right half shows the 6-object regime. For visual clarity, all objects are rendered at a higher resolution here than in the training input.

One property of our model is the inherent presence of noise from the visual encoder. In contrast to state-to-state models such as the Interaction Net, here the dynamic predictor's input is inherently noisy due to the discretization of our synthetic dataset rendering. Surprisingly, this noise seemed to confer an advantage because it helped the model learn to overcome temporally compounding errors generated by inaccurate predictions. This is especially notable when doing long term roll outs where we achieve performance which surpasses even a pure state-to-state Interaction Net. Since this dependence on noise would be inherent in any model operating on visual input, we postulate that this is an important feature of *any* prediction model and warrants further research.

While experimentation with variable number of objects falls outside the scope of the material presented here, this is an important direction that could be explored in further work. Importantly, INs generalize out of the box to scenes with a variable number of objects. Should the present form of the perceptual encoder be insufficient to support this type of generalization, this could be addressed by using an attentional encoder and order-agnostic loss function.

Our Visual Interaction Network provides a step toward understanding how representations of objects, relations, and physics can be learned from raw data. This is part of a broader effort toward understanding how perceptual models support physical predictions and how the structure of the physical world influences our representations of sensory input, which will help AI research better capture the powerful object- and relation-based system of reasoning that supports humans' powerful and flexible general intelligence.

**Acknowledgments**

We thank Max Jaderberg, David Reichert, Daan Wierstra, and Koray Kavukcuoglu for helpful discussions and insights.

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
