[Supplementary Material]

# Visual Interaction Networks: Learning a Physics Simulator from Video
# Supplementary Material

**Nicholas Watters, Andrea Tacchetti, Théophane Weber**

**Razvan Pascanu, Peter Battaglia, Daniel Zoran**

DeepMind

London, United Kingdom

{nwatters, atacchet, theophane,
razp, peterbattaglia, danielzoran}@google.com

## 1  Supplementary Videos

We provide videos showing sample VIN rollout sequences and dataset examples. In all videos, for visual clarity the objects are rendered at a higher resolution than they are in the input data.

### 1.1  Sample VIN Rollout Videos

See the videos at

https://goo.gl/RjE3ey

These show rendered VIN rollout position predictions compared to the ground-truth (unobserved) system simulation for 3-object and 6-object datasets of all force systems. In each video, the VIN rollout is on the right-hand side and the ground-truth on the left-hand side. Rollouts are 200 steps long for 3-object systems and 100 steps long for 6-object systems (except for Drift, where we use 35 steps for all rollouts to ensure objects don't drift out of view). The VIN rollout tracks the ground truth quite closely for most datasets. Two rollout examples are provided for each dataset.

### 1.2  Dataset Examples

See the videos at

https://goo.gl/yVQbUa

These show examples of the 3-object and 6-object datasets of all force systems. The VIN and all baselines receive 6 frames of these as input during training and rollouts.

## 2  Training

### 2.1  Loss Function

Given a state code containing a representation of position and velocity for each object in the scene, we compute the loss as follows: First, apply the decoder to each slot in the state code (there is one slot for each object) to get a stack of 4-vectors, one for each object. Then, take the mean squared error of each 4-vector slot with respect to the ground truth positions and velocity, and average over slots. This is a loss term for a state code. Since our models have temporal offsets up to length 4, we have 4 encoded state codes. For each of these encoded state codes, we take the mean of the 4 corresponding state code losses to obtain the encoding loss. For the predicted (rollout) state codes, we

take the weighted average of over the 8 predicted state codes of the state code losses (the weighting is described in the main text). The final loss is the sum of the encoding loss and the prediction loss.

Note that for the VIN without temporal offsets, the encoding loss has 1 term instead of 4.

## 2.2 Training Parameters

We use the following training parameters for all models:

- Training steps: $5 \cdot 10^5$
- Batch Size: 4
- Gradient Descent Optimizer: Adam, learning rate $5 \cdot 10^{-4} e^{-t/\alpha}$ where $\alpha = 1.5 \cdot 10^5$ and $t$ is the training step.
- Rollout frame temporal discount (factor by which future frames are weighted less in the loss): $1 - \gamma$ with $\gamma = e^{-t/\beta}$ where $\beta = 2.5 \cdot 10^4$ and $t$ is the training step.

# 3 VIN Model Details

In all models, all activation functions were ReLU. All network variables were initialized with zero bias and weights drawn from a normal distribution with standard deviation $(input\ size)^{-1/2}$ (for convolutional weight variables, $input\ size$ is the fan-in size).

## 3.1 Visual Encoder

See the main text for schematics and high-level summary descriptions of all model components. Here we describe parameters and details.

The **visual encoder** takes a sequence of three images as input and outputs a state code. Its sequence of operations on frames $[F_1, F_2, F_3]$ is as follows:

- Apply an image pair encoder (described below) to $[F_1, F_2]$ and $[F_2, F_3]$, obtaining $S_1$ and $S_2$. These are length-32 vectors.
- Apply a shared linear layer to convert $S_1$ and $S_2$ to tensors of shape $N_{object} \times 64$. Here $N_{object}$ is the number of objects in the scene, and 64 is the length of each state code slot.
- Concatenate $S_1$ and $S_2$ in a slot-wise manner, obtaining a single tensor $S$ of shape $N_{object} \times 128$.
- Apply a shared MLP with one hidden layer of size $64$ and a length-64 output layer to each slot of $S$. The result is the encoded state code.

The Image Pair Encoder takes two images as input and outputs a candidate state code. Its sequence of operations on frames $[F_1, F_2]$ is as follows:

- Stack $F_1$ and $F_2$ along their color-channel dimension.
- Independently apply two 2-layer convolutional nets, one with kernel size 10 and 4 channels and the other with kernel size 3 and 16 channels. Both are padded to preserve the input size. Stack the outputs of these convolutions along the channel dimension.
- Apply a 2-layer size-preserving convolutional net with 16 channels and kernel-size 3.
- Inject two constant coordinate channels, representing the x- and y-coordinates of the feature matrix. These two channels are a meshgrid with min value 0 and max value 1.
- Convolve to unit height and width with alternating convolutional and $2 \times 2$ max-pooling layers. The convolutional layers are size-preserving and have kernel size 3. In total, there are 5 each of convolutional and max-pooling layers. The first three layers have 16 channels, and the last two have 32 channels. Flatten the result into a 32-length vector. This is the image pair encoder's output.

## 3.2 Dynamics Predictor

The **dynamics predictor** takes a sequence of 4 consecutive state codes $[S_1, ..., S_4]$ and outputs a predicted state code $S^{pred}$, as follows:

- Temporal offsets are $1, 2, 4$, so we have IN cores $C_1, C_2, C_4$. Since the temporal offset indexing goes back in time, we apply $C_4$ to $S_1$, $C_2$ to $S_3$, and $C_1$ to $S_4$. Let $S_1^{candidate}, S_3^{candidate}, S_4^{candidate}$ denote the outputs.
- Apply a shared slot-wise MLP aggregator with sizes $[32, 64]$ to the concatenation of $S_i^{1,3,4}$ for each $i \in \{1, ..., N_{object}\}$. The resulting state code is the dynamics predictor's output.

The **Interaction Net** core takes a state code $[M_i]_{1 \le i \le N_{object}}$ as input and outputs a candidate state code, as follows:

- Apply a Self-Dynamics MLP with sizes $[64, 64]$ to each slot $M_i$. Let $[M_i^{self}]_{1 \le i \le N_{object}}$ denote these.
- Apply a Relation MLP with sizes $[64, 64, 64]$ to the concatenation of each pair of distinct slots. Let $[M_{ij}^{rel}]_{1 \le i \ne j \le N_{object}}$ denote the outputs.
- Sum for each slot the quantities computed so far, to produce an updated slot. Specifically, let $M_i^{update} = M_i^{self} + \sum_j M_{ij}^{rel}$.
- Apply an Affector MLP with sizes $[64, 64, 64]$ to each $M_i^{update}$, yielding $M_i^{affect}$.
- For each slot, apply a shared MLP with sizes $[32, 64]$ to the concatenation of $M_i$ and $M_i^{affect}$. The resulting state code is the output.

The **IN from State** model uses the same dynamics predictor, but no encoder (it is given the position/velocity vectors for directly).

## 3.3 RNN Models

The **Visual RNN** model uses the same parameters for the visual encoder as the VIN. The RNN dynamics predictor has a single MLP core with sizes $[64, 64, 64, 64, 64]$ and the same temporal offset of $\{1, 2, 4\}$ and shared slot-wise MLP aggregator with sizes $[32, 64]$ as the VIN.

## 3.4 LSTM

The **Visual LSTM** model uses the same parameters for the visual encoder as the VIN. The LSTM dynamics predictor has a single LSTM/MLP core consisting of a pre-processor MLP with sizes $[64, 64]$, an LSTM with $128$ hidden units, and a post-processor MLP with sizes $32, 32$. This is the core of an temporal offset-aggregating MLP with sizes $[32, 64]$ and temporal offsets $1, 2, 4$.

The **LSTM from State** model uses the LSTM dynamics predictor on position/velocity states.

# 4 Importance of Temporal Offsets

In the main text, we highlighted the importance of the relation network in the model's interaction net by showing results from an ablation experiment with that network removed. Here, we highlight the importance of temporal offset aggregation by showing results from an ablation experiment without any temporal offset aggregation. Specifically, we give the dynamics predictor only one core $C_1$ and only one single frame of input $S_1$ (see SM Secion 3.2). Everything else remains the same, including the shared slot-wise MLP aggregator, which now takes not a concatenation over cores but the slots from just the one core $C_1$. Additionally, note that the auxiliary encoding loss now has only one term, instead of four.

Figure 1 shows the results of this experiment in a format analogous to Figure 3 of the main text. The only difference here is that we replace the relation net ablation experiment results with the temporal offset ablation experiment results described here. We see that the performance of the VIN without temporal offsets is significantly lower than that of the VIN, hence temporal offset aggregation is a crucial component of the model.

Figure 1: **Importance of Temporal Offsets.** Model comparison of Inverse Normalized Loss. This includes results from the VIN without temporal offsets (yellow). Results for all other models are the same as in Figure 3 of the main text. The poor performance of the VIN without temporal offsets shows that temporal offset aggregation is critical for the performance of the VIN model.

## 5 Datasets

### 5.1 Physical Systems

We simulated each physical system with Newton's Method and internal simulation timestep small enough that there was no visual distinction after 300 frames when using the RK4 method. We use the specific force laws below:

- **Spring** A pair of objects at positions $\vec{p}_i$ and $\vec{p}_i$ obey Hooke's law

$$\vec{F}_{ij} = -\kappa \vec{d}_{ij} - \varepsilon \frac{\vec{d}_{ij}}{|\vec{d}_{ij}|}$$

  where $F_{ij}$ is the force component on object $j$ from object $i$. Here $\vec{d}_{ij} = \vec{p}_i - \vec{p}_j$ is the displacement between the objects, $\kappa$ is the spring constant, and $\varepsilon$ is the equilibrium. We use $\varepsilon = 0.45$

- **Gravity** A the pair of objects with masses $m_i$, $m_j$ obey Newton's Law

$$\vec{F}_{ij} = -G \frac{m_i m_j \vec{d}_{ij}}{|\vec{d}_{ij}|^3}$$

  where $G$ is the gravitational constant. In practice, we upper-bounded the gravitational force to avoid instability due to the "slingshot" effect when two objects pass extremely close to each other. To further prevent objects from drifting out of view, we also applied a weak attraction towards the center of the field of view. The system effectually operates within a parabolic bowl.

- **Billiards** A pair of balls only interact when they touch, in which case they bounce off of each other instantaneously and with total elasticity. The bounces conserve kinetic energy and total momentum, as if the objects are perfect billiard balls. In addition, the balls bounce off the edges of the field of view.

- **Magnetic Billiards** A pair of objects with charges $q_i$, $q_j$ obey Coulomb's law

$$F_{ij} = k \frac{q_1 q_2 \vec{d}_{ij}}{|\vec{d}_{ij}|^3}$$

  where $k$ is Coulomb's constant. In addition, the balls bounce off the edges of the field of view as in the Billiards system.

|  | pixel-to-state models | | | state-to-state models | |
| --- | --- | --- | --- | --- | --- |
| 3-object datasets | VIN | Visual LSTM | Visual RNN | VIN from State | LSTM from State |
| Spring | 1.646 | 1.831 | 3.272 | 0.426 | 1.844 |
| Gravity | 1.194 | 1.288 | 1.572 | 0.146 | 0.191 |
| Magnetic Billiards | 1.792 | 1.878 | 2.911 | 0.454 | 1.863 |
| Billiards | 1.391 | 1.600 | 2.752 | 0.942 | 2.507 |
| Drift | 2.474 | 2.920 | 3.663 | 0.0017 | 0.0052 |
|  |  |  |  |  |  |
| 6-object datasets | VIN | Visual LSTM | Visual RNN | VIN from State | LSTM from State |
| Spring | 0.565 | 0.608 | 0.858 | 0.235 | 0.324 |
| Gravity | 0.416 | 0.422 | 0.597 | 0.092 | 0.157 |
| Magnetic Billiards | 0.750 | 0.836 | 1.374 | 0.349 | 0.791 |
| Billiards | 0.918 | 1.022 | 2.582 | 0.817 | 1.919 |
| Drift | 0.749 | 0.831 | 1.083 | 0.0025 | 0.0069 |

Table 1: **Mean Euclidean Prediction Error On 8-Step Rollouts.** These values show the Mean Euclidean Prediction Error on the length-8 test rollouts. All values are scaled by 100, so they show a geometric deviation as a percentage of the frame width. Our model outperforms all pixel-to-state baselines on all datsets, and also outperforms the LSTM state-to-state baseline on some datasets. We believe the unexpectedly high values on Drift result from objects nearly drifting out of view before simulations are terminated. We also believe the lower values on 6-object datasets is a function of the slower velocity of those datasets.

- **Drift** In this system there are no forces, so the objects simply drift with their initial velocity. We terminate all simulations before the objects completely exit the frame, though bound the initial positions and velocities so that this never occurs before 32 timesteps.

Unspecified parameters $\kappa$, $G$, and $k$ were tuned with the render stride for each dataset and object number to make the object velocities look reasonable.

We initialize each object's initial position uniformly within a centered box of width $0.8$ times the framewidth. We initialize each object's velocity uniformly at random, except for Gravity, where we initialize each object's velocity as the counter-clockwise vector tangent with respect to the center of the frame, then add a small random vector (this was necessary to ensure stability of the system).

For the unbounded systems (Gravity, Springs, and Drift), after the velocities are initialized we enforce net zero momentum by subtracting an appropriate vector from each ball's initial velocity. For these systems we also center the objects' positions so that the center of mass lies in the center of the frame. These measures ensure the entire system remains in view.

For all systems except Drift we apply a weak frictional force (linearly proportional to each ball's area), to ensure that any accumulation of numerical inaccuracies does not cause instability in any systems, even after many hundreds of timesteps.

We render each system as a $32 \times 32$ RGB video in front of a CIFAR10 natural image background. For systems that allow occlusion (every system except Billiards), we use a foreground/background ordering of the balls by color, and this ordering is fixed for the entire dataset.

## 5.2 Numerical Results

In Tables 1 and 2 we show values of the Mean Euclidean Prediction error on all models and all datasets after 8 and 50 rollout steps, respectively. These values numerically represent time-slices of Figure 6 in the main text, partitioned by dataset.

| | Euclidean deviation after 50 simulation timesteps | | | | |
|---|---|---|---|---|---|
| | Our Model | Visual LSTM | Visual RNN | Our Predictor | LSTM Predictor |
| Spring | 0.046 | 0.249 | 0.157 | 0.063 | 0.324 |
| Gravity | 0.008 | 0.048 | 0.043 | 0.013 | 0.081 |
| Magnetic Billiards | 0.111 | 0.398 | 0.314 | 0.179 | 0.332 |
| Billiards | 0.151 | 0.391 | 0.308 | 0.199 | 0.348 |
| | Euclidean deviation per object after full frame width is travelled | | | | |
| | Our Model | Visual LSTM | Visual RNN | Our Predictor | LSTM Predictor |
| Spring | 0.069 | 0.304 | 0.213 | 0.091 | 0.360 |
| Gravity | 0.009 | 0.038 | 0.038 | 0.010 | 0.057 |
| Magnetic Billiards | 0.118 | 0.417 | 0.455 | 0.165 | 0.354 |
| Billiards | 0.179 | 0.470 | 0.395 | 0.223 | 0.411 |

Table 2: **Mean Euclidean Prediction Error for 50-Step Rollouts.** These values show the Mean Euclidean Prediction Error on length-50 test rollouts. All values are scaled by 100, so they show a geometric deviation as a percentage of the frame width. Our model out-performs all other models, including state-to-state models.