[Reviews · NeurIPS 2017]

Reviewer 1



Visual Interaction Networks. This paper presents a general-purpose model based on convolutional networks and recurrent neural networks with an interaction network to predict future states from raw visual observations. The use case is compelling and the models demonstrate excellent performance in predicting future states and also demonstrating good performance with noisy backgrounds and springs with invisibility and springs and billiards with variable mass. MAJOR - in the section training procedure 3.3, please clarify the sequence of 8 unseen future states. I read from 3.1 that the training sequence is 14 frames. Are these next 8 unseen future state frames the next 8 frames after the training sequence of 14 frames or are these randomly selected frames from future time? - apologies but the actual setup of the tests is unclear to me from the descriptions. the model is trained on the training simulations. then each test simulation is fed into the model and the inverse normalized loss is calculated for the vision with ground truth dynamics and the test of loss of the model? - I found it curious that the authors didn't compare to some other techniques in the related works section. The authors clearly show the merits of their model in the comparisons that they did by altering aspects of the network architecture. It may be correct to say that there is no prior work to compare to and would be an excellent statement to make if that is true. I apologize that I don't know enough about this area of research to know what other approaches would be reasonable to compare (if any). - Is the loss calculated in the inverse normalized loss the same or different than the loss in section 3.3? MINOR comment - I would close the related work section with a paragraph summing up the differences of this work from previous work. - are frames from each simulation the same as timesteps as discussed in table 1? Comments in categories: Quality The paper seems technically sounds. The claims are well-supported by experimental results. The work is complete. There is limited discussion of the strengths and weaknesses of the approach especially as there is no comparison to prior work. It is possible that this is completely novel and I would recommend the authors state that if it is true. It is very compelling to say so. Clarity The paper is clearly written for the most part and well-organized. There are some points of clarification that commented above. Originality I am not qualified to put this work into context. The authors do reference prior work but no results from these experiments are compared as baselines. The difficulty of the problem and the need for the additional architecture in the model is warranted through the comparison to the visual RNN in figure 3. Significance The results seem important if not at least performing close to the ground truth. I think other practitioners would build upon these ideas. Prior work is not demonstrated in their experiments so it is impossible to assess their work in context though this work may be completely novel and I would like to see a statement along these lines. Closing the related work with a comment of how their work fits in would be greatly warranted to the uninitiated in this area. In general, the authors present compelling evidence that their model predicts future state from raw visual observations.

Reviewer 2



The paper presents an approach with predicts the next few frames given a sequence of input image frames. It introduces a Visual Interaction Neural net model to perform the prediction task. The model includes 3 different components, CNN, a dynamic predictor and state decoder. The proposed approach is able to outperform the proposed baseline algorithms with respect to loss and Euclidean prediction error. The problem is very interesting but the paper is not self-explanatory as it relies on supplementary document significantly which makes it almost impossible to understand based on the content in the paper. As a person who actively work with deep learning techniques, it was really hard for me to understand the dynamic predictor structure based on the paper. The interaction net is very new to me, and I assume to most of NIPS crew too, and hence I expect to hear more about it in the paper. Perhaps, you can buy some spaces by removing some rows from Table 1 and explain the details of your model more, especially interaction network. Another problem is elaborating the loss function in the paper, how do you calculate it? It is not clear to me. Looks like the interaction networks understand the underlying physics between the objects very well, is this correct? And if yes, how could it be extended to other scenarios with more complexity (such as video games). Have you ever tried it? How does the proposed normalized weighted sum(line1791) differs from the general loss calculation(same weight for all frame) in your experiments? Intuitively, it makes sense to use the normalized one but I am interested to know the impact on the result. 1-2 line explanation could be enough.

Reviewer 3



Summary --- Interaction Networks (INs) [2] and the Neural Physics Engine [7] are two recent and successful instantiations of a physics engine. In each case a hand programed 2d physics engine was used to simulate dynamics of 2d balls that might bounce off each other like billiards, orbit around a made up gravitational well, or a number of other settings. Given ball states (positions and velocities), these neural approaches predict states at subsequent time steps and thus simulate simple entity interaction. This paper continues in the same vein of research but rather than taking states as input, the proposed architecture takes images directly. As such, the proposed Visual Interaction Networks (VINs) must learn to embed images into a space which captures state information and learn how to simulate 2d physics in that embedding space. The VIN comes in three parts. 1) The visual encoder is a CNN which takes a sequence of frames which show the three most recent states of a simulation and embeds them into a different embedding for each object (the number of objects is fixed). Some tricks help the CNN focus on pairs of adjacent frames and do position computations more easily. 2) The dynamics predictor does the same job as an IN but takes state embeddings as input (from the visual encoder) and predicts state embeddings. It's implemented as a triplet of INs which operate at different time scales and are combined via an MLP. 3) The state decoder is a simple MLP which decodes state embeddings into hand coded state vectors (positions and velocities). Hand coded states are be extracted from the ground truth 2d simulation and a loss is computed which forces decoded state embeddings to align to ground truth states during training, thus allowing the dynamics predictor to learn an appropriate 2d physics model. This model is tested across a range of dynamics similar to that used in INs. Performance is reported both in terms of MSE (how close are state predictions to ground truth states) and relative to a baseline which only needs to extract states from images but not predict dynamics. Across different number of objects and differing dynamics VINs outperform all baselines except in one setting. In the drift setting, where objects do not interact, VINs are equivalent to an ablated version that removes the relational part from each IN used in the dynamics predictor. When compared to the simulation rolled out by the ground truth simulator using MSE all models show increased errors longer from the initial state (as expected), but VIN is smallest at every time step. A particularly interesting finding is that VINs even outperform INs which are given ground truth states and need not decode them from images. This is hypothesized to be due to noise from the visual component of VINs. Finally, videos of VIN rollouts are provided and appear qualitatively to convincingly simulate each intended 2d dynamics. Strengths --- This is a well written paper which presents the next step in the line of work on neural physics simulators, evaluates the proposed method somewhat thoroughly, and produces clean and convincing results. The presentation also appropriately orients the work relative to old and new neural physics simulators. Weaknesses --- This paper is very clean, so I mainly have nits to pick and suggestions for material that would be interesting to see. In roughly decreasing order of importance: 1. A seemingly important novel feature of the model is the use of multiple INs at different speeds in the dynamics predictor. This design choice is not ablated. How important is the added complexity? Will one IN do? 2. Section 4.2: To what extent should long term rollouts be predictable? After a certain amount of time it seems MSE becomes meaningless because too many small errors have accumulated. This is a subtle point that could mislead readers who see relatively large MSEs in figure 4, so perhaps a discussion should be added in section 4.2. 3. The images used in this paper sample have randomly sampled CIFAR images as backgrounds to make the task harder. While more difficult tasks are more interesting modulo all other factors of interest, this choice is not well motivated. Why is this particular dimension of difficulty interesting? 4. line 232: This hypothesis could be specified a bit more clearly. How do noisy rollouts contribute to lower rollout error? 5. Are the learned object state embeddings interpretable in any way before decoding? 6. It may be beneficial to spend more time discussing model limitations and other dimensions of generalization. Some suggestions: * The number of entities is fixed and it's not clear how to generalize a model to different numbers of entities (e.g., as shown in figure 3 of INs). * How many different kinds of physical interaction can be in one simulation? * How sensitive is the visual encoder to shorter/longer sequence lengths? Does the model deal well with different frame rates? Preliminary Evaluation --- Clear accept. The only thing which I feel is really missing is the first point in the weaknesses section, but its lack would not merit rejection.